# On the Utility of Equivariance and Symmetry Breaking in Deep Learning Architectures on Point Clouds

## Abstract

This paper explores the key factors that influence the performance of models working with point clouds, across different tasks of varying geometric complexity. In this work, we explore the trade-offs between flexibility and weight-sharing introduced by equivariant layers, assessing when equivariance boosts or detracts from performance. It is often argued that providing more information as input improves a model's performance. However, if this additional information breaks certain properties, such as SE(3) equivariance, does it remain beneficial? We identify the key aspects of equivariant and non-equivariant architectures that drive success in different tasks by benchmarking them on segmentation, regression, and generation tasks across multiple datasets with increasing complexity. We observe a positive impact of equivariance, which becomes more pronounced with increasing task complexity, even when strict equivariance is not required.

## 1 Introduction

The inductive bias of weight sharing in convolutions, as introduced in LeCun et al. (2010) traditionally refers to applying the same convolution kernel (a linear transformation) across all neighborhoods of an image. To extend this to transformations beyond translations, Cohen & Welling (2016) introduced Group Equivariant CNN (G-CNNs), adding group equivariance properties to encompass group actions and have weight-sharing across group convolution kernels. G-CNN layers are explicitly designed to maintain equivariance under group transformations, allowing the model to handle transformations naturally without needing to learn invariance to changes that preserve object identity. Following this work—in the spirit of 'convolution is all you need' (Cohen et al., 2019), several works emerged like (Ravanbakhsh et al., 2017; Worrall et al., 2017; Kondor & Trivedi, 2018; Bekkers et al., 2018; Weiler et al., 2018; Cohen et al., 2019; Weiler & Cesa, 2019; Bekkers, 2019; Sosnovik et al., 2019; Finzi et al., 2020). Advancements in geometric deep learning have demonstrated the effectiveness of incorporating geometric structure as an inductive bias, which reduces model complexity while enhancing generalization and performance (Bronstein et al., 2021). Thus, incorporating group structure into neural networks has become a promising area of research.

However, there is a growing debate as to whether group structure is overly restrictive and if similar advantages could be obtained by simply adding more data. To address this question, we thoroughly investigate the impact of equivariant versus non-equivariant layers in various computational tasks. We explore the balance between leveraging group structures that can provide powerful inductive biases and maintaining the inherent model flexibility. We implemented a convolutional architecture in which the linear layers are either classical or group convolution layers, whilst the overall architecture remains otherwise identical for fair comparison. We evaluate this model on tasks of varying complexity and the extent to which equivariance is desirable under the task description. Our goal is to understand how these design choices affect performance, generalization capabilities, and computational efficiency.

The contributions of this paper can be summarized as follows:

- **Hypothesis formulation**: We present a set of hypotheses that allow us to investigate the different aspects of equivariant neural networks and symmetry breaking.

- **Empirical study**: To test these hypotheses, we conduct experiments on the point cloud datasets Shapenet 3D, QM9, and CMU Motion Capture across different tasks to assess the empirical effects of using equivariant versus non-equivariant layers.

- **Scalable architecture**: We present a scalable architecture, Rapidash, based on recent work on regular group convolutional architectures, that enables fast computation of different group equivariant and non-equivariant methods, facilitating comprehensive testing of our hypotheses.

## 2 BACKGROUND

In this section, we begin by explaining how equivariant neural networks differ from non-equivariant ones, focusing on their architectural distinctions and the impact of weight-sharing—or the lack thereof—on data efficiency. We then delve into the specifics of 3D convolutions and separable group convolutions, which inform the design of the architecture presented later.

### 2.1 ARGUMENTS FOR AND AGAINST EQUIVARIANCE

Equivariant neural networks differ from standard networks in four key aspects:

1. **High-dimensional representation spaces**: They typically represent data on higher-dimensional feature maps, allowing for richer internal representations.

2. **Constraint layers**: They employ constrained linear layers that are equivariant to specific transformations, which, while less flexible, preserve important structural properties of data.

3. **Weight-sharing**: These constraints not only prevent overfitting to nuisance variables (e.g. arbitrary rotations) but also introduce a form of weight-sharing that could benefit learning.

4. **Data efficiency**: Equivariant architectures are data efficient, as they do not need to learn patterns repeatedly for different transformations (poses) under which they may appear.

When comparing equivariant networks such as group convolutional CNNs (GCNNs) to standard CNNs, we observe that: (1) the higher dimensionality could possibly be offset in a standard CNN by increasing hidden dimensions; (2) the equivariance constraint may disadvantage equivariant methods in settings where such constraints are not crucial; however, (3) the induced weight-sharing and (4) improved data efficiency might compensate for these limitations by enabling more effective and efficient learning. So, there are arguments against (1-2) and in favor of G-CNNs (3-4).

This paper explores the scenarios in which G-CNNs outperform standard CNNs. We hypothesize that equivariant networks offer advantages not only in tasks explicitly requiring equivariance but also in challenging tasks where strict equivariance is not necessary.

### 2.2 INTRODUCTION TO EQUIVARIANT NEURAL NETWORKS

To address the above arguments we first provide a high-level introduction to group equivariant convolutions in this section and provide the technical details in the next Section 2.3.

**Linear Layers and Equivariance** Consider a vanilla neural network of the form:

$$\text{NN}(x) = \left[ \sigma^{(L)} \circ L^{(L)} \circ \ldots \sigma^{(2)} \circ L^{(2)} \circ \sigma^{(1)} \circ L^{(1)} \right](x), \tag{1}$$

where $L^{(l)}$ are linear layers and $\sigma^{(l)}$ are element-wise activation functions. The input $x$ can represent various data structures, such as images in $\mathbb{R}^{X \times Y \times C}$ or graphs in $\mathbb{R}^{N \times C}$. In such an architecture—as well as in modern variants, the bulk of the computations are done through linear transformations $L$, which form features as linear combinations of input patterns. For structured data, these linear transformations are designed to be *equivariant* to preserve data structure. For instance, graph networks require permutation equivariance, while image processing networks demand translation equivariance. These constraints result in *convolution operators* which are efficiently implementable via sparse and parallelized operations.

**Equivariant Layer Design** G-CNNs are equivariant networks that operate over higher-dimensional feature maps in which an additional axis (fiber) is used to keep track of feature information relative to (sub-)groups of transformations. For example, *regular* roto-translation equivariant group convolutions for image data add an extra axis for storing the response to rotated versions of a convolution kernel (Cohen & Welling, 2016). In other words, a standard 2D CNN uses linear maps $L : \mathbb{R}^{X \times Y \times C} \to \mathbb{R}^{X \times Y \times C'}$, where $C$ and $C'$ are the in- and output channel dimensions, $X, Y$ the spatial dimensions. A G-CNN internally uses linear layers $L : \mathbb{R}^{X \times Y \times F \times C} \to \mathbb{R}^{X \times Y \times F \times C'}$, in which $F$ is the number of grid points (e.g. discrete rotations) on which the feature maps are sampled.

**Hidden Representations and Constraints** While in principle increasing the dimensionality of hidden representations can enhance capacity in both standard and equivariant networks, the key distinction lies in the constraints imposed on the linear layers. Equivariant layers, despite potentially having similar total dimensionality, are more constrained in their operations. This constraint, while limiting flexibility, introduces beneficial properties such as weight-sharing and invariance to certain transformations.

For instance, an unconstrained layer $L : \mathbb{R}^{X \times Y \times FC} \to \mathbb{R}^{X \times Y \times FC'}$ may be more expressive than an equivariant layer $L : \mathbb{R}^{X \times Y \times F \times C} \to \mathbb{R}^{X \times Y \times F \times C'}$, even when the total dimensionality of the feature maps are equivalent. However, the equivariant layer's constraints can lead to improved generalization and sample efficiency in certain tasks. Our study aims to identify the conditions under which the benefits of equivariance outweigh the potential limitations in expressivity, particularly in tasks where strict equivariance is not explicitly required.

**Equivariant Networks and Symmetry-Breaking** Relaxing equivariant constraints to improve generalization has been well studied. Finzi et al. (2021) presents RPP, which allows for relaxing architectural constraints of equivariance by introducing neural network priors that allow for approximate equivariance. Prior works like Wang et al. (2022); van der Ouderaa et al. (2022); Pertigkiozoglou et al. (2024) focus on relaxing equivariant constraints and show the benefit in performance on learning approximate or partial equivariance. To understand the effects of explicit symmetry breaking while learning strict equivariance in tasks with different geometric complexity, our approach consists of breaking a) internal symmetry and b) external symmetry. See A.3 for more details.

**Data Efficiency and Large-Scale Learning** The data efficiency provided by equivariant architectures is a significant advantage, particularly in scenarios with limited data. By leveraging symmetries in the data, these networks can learn patterns once and recognize them across various transformations, reducing the amount of data required for effective learning. However, as datasets grow larger, a critical question emerges: Does the benefit of equivariance diminish when data is abundant? We hypothesize that equivariant methods maintain their relevance even in large-scale learning scenarios for several reasons:

1. **Structured Representation Learning:** The constraints imposed by equivariance guide the network to learn more structured and potentially more meaningful representations, which may generalize better regardless of dataset size.
2. **Computational Efficiency:** Equivariant networks can potentially learn from large datasets more efficiently, requiring fewer parameters and computations to achieve similar or better performance.
3. **Inductive Bias:** The equivariance constraint serves as a strong inductive bias, which may lead to better generalization even when data is plentiful, by focusing the model on learning relevant features rather than spurious correlations.

Our study aims to (empirically) investigate these hypotheses (precise formulation in Section 3.1), comparing the performance and learning dynamics of equivariant networks against standard architectures across various dataset sizes and task complexities. By doing so, we seek to delineate the conditions under which equivariance provides substantial benefits, even in the era of big data.

## 2.3 EQUIVARIANT LINEAR LAYERS (CONVOLUTIONS)

**3D Point Cloud Convolutions** Let us consider a 3D point cloud $\mathcal{X} = \{\mathbf{x}_1, \mathbf{x}_2, \ldots, \mathbf{x}_N\} \subset \mathbb{R}^3$ of $N$ points and assume feature fields $f : \mathcal{X} \to \mathbb{R}^C$. Thus with every point $\mathbf{x}_i$ we have an associated

$C$-dimensional feature vector $f(\mathbf{x}_i) \in \mathbb{R}^C$. We further assume connectivity between points to be given in the form of neighborhood sets, that is, let $\mathcal{N}(i) \subset \mathcal{V}$ denote a subset of nodes connected to node $i$, with $\mathcal{V} = \{1, 2, \ldots, N\}$ indicating the set that indexes the point cloud.

The general form of a linear layer for such point cloud feature fields is given by (Bekkers, 2019)

$$[Lf](\mathbf{x}_i) = \sum_{j \in \mathcal{N}(i)} k(\mathbf{x}_i, \mathbf{x}_j) f(\mathbf{x}_j) \,, \tag{2}$$

where we note that $k(\mathbf{x}_i, \mathbf{x}_j) \in \mathbb{R}^{C'} \times \mathbb{R}^C$ is a matrix depending on both the receiving and sending nodes, $i$ and $j$ respectively, such that the output feature map has $C'$ channels. Also, note that the aggregation is permutation invariant. In works such as (Cohen et al., 2019; Bekkers, 2019) it is shown that when constraining linear layers of the form (2) to be equivariant, they become (group) convolutions. For example, if we want (2) to be translation equivariant, it *has to take the form*

$$[Lf](\mathbf{x}_i) = \sum_{j \in \mathcal{N}(i)} k(\mathbf{x}_j - \mathbf{x}_i) f(\mathbf{x}_j) \,. \tag{3}$$

I.e., $k(\mathbf{x}_i, \mathbf{x}_j)$ must then be constrained to be a one-argument kernel $k(\mathbf{x}_j - \mathbf{x}_i)$, *conditioned on relative position*. If we further want the linear layer to be equivariant to both translations *and* rotations, the kernel is further constrained to be symmetric via

$$[Lf](\mathbf{x}_i) = \sum_{j \in \mathcal{N}(i)} k(\|\mathbf{x}_j - \mathbf{x}_i\|) f(\mathbf{x}_j) \,. \tag{4}$$

I.e., the kernel can only depend on *pair-wise distances*. The influential works Schnett (Schütt et al., 2023) and PointConv (Wu et al., 2019) are of this type.

**Group convolutions**   In case one does not want to impose any further constraints on the kernel $k$ but still wants to remain fully SE(3) equivariant, one has no other option than adding an axis over which to organize convolution responses in terms of rotations (Bekkers, 2019, Theorem 1). One then has to utilize lifting convolutions, followed by group convolutions:

$$\text{lifting conv:} \qquad [Lf](\mathbf{x}_i, \mathbf{R}) = \sum_{j \in \mathcal{N}(i)} k(\mathbf{R}^T(\mathbf{x}_j - \mathbf{x}_i)) f(\mathbf{x}_j) \,, \tag{5}$$

$$\text{group conv:} \qquad [Lf](\mathbf{x}_i, \mathbf{R}) = \sum_{j \in \mathcal{N}(i)} \int_{SO(d)} k(\mathbf{R}^T(\mathbf{x}_j - \mathbf{x}_i), \mathbf{R}^T\tilde{\mathbf{R}}) f(\mathbf{x}_j, \tilde{\mathbf{R}}) \mathrm{d}\tilde{\mathbf{R}} \,, \tag{6}$$

with $\mathrm{d}\tilde{\mathbf{R}}$ denoting the Haar measure over the rotation group $SO(3)$. Note that now the convolution kernel in equation 5 is an unconstrained function over $\mathbb{R}^3$, and that of Eq. 6 over $\mathbb{R}^3 \times SO(3)$, and that this kernel is rotated for every possible $\mathbf{R} \in SO(3)$. Both the lifting and group convolution layers generate feature maps $\mathcal{X} \times SO(3) \to \mathbb{R}^{C'}$ over the joint space of positions $\mathcal{X} \subset \mathbb{R}^3$ and rotations $SO(3)$. In other words, at each point $\mathbf{x}_i$ we now have a signal over the rotation group $SO(3)$ which stores the convolution response for every "pose" $\mathbf{R}$ in $SO(3)$, and the linear layer is defined using convolution kernels that match feature patterns of relative spatial and rotational poses.

Eqs. 5 and 6 are forms of *regular* group convolutions Cohen & Welling (2016). Brandstetter et al. show that such layers can also be implemented through tensor field layers (Thomas et al., 2018), which form a popular class of *steerable* group convolutions that are parametrized by Clebsch-Gordan tensor products and work with vector fields that transform via irreducible representations (Weiler et al., 2021). Following Bekkers et al. (2024), we note however, that tensor-field networks unnecessarily constrain neural network design—as they require specialized activation functions, they require in-depth knowledge of representation theory, and they are computationally demanding due to the use of Glebsch-Gordan tensor products. Our study therefore focuses on regular group convolutions.

**Separable group convolution**   Regular group convolutions can be efficiently computed when the kernel is factorized via

$$k_{c'c}(\mathbf{x}, \mathbf{R}) = k_c^{\mathbb{R}^3}(\mathbf{x}) k_c^{SO(3)}(\mathbf{R}) k_{c'c}^{(channel)} \,,$$

with $c, c'$ the row and column indices of the "channel mixing" matrix. Then Eq. 6 can be split into three steps that are each efficient to compute: a spatial interaction layer (message passing), a point-wise SO(3) convolution, and a point-wise linear layer (Knigge et al., 2022; Kuipers & Bekkers, 2023). It would result in the group convolutional counterpart of *depth-wise separable convolution* (Chollet, 2017), which separates convolution in two steps (spatial mixing and channel mixing).

**Compute and memory-efficient group convolutions** Finally, we base our equivariant layers on recent work (Bekkers et al., 2024) that defines separable SE(3) group convolutions over the space $\mathcal{X} \times S^2$, thus working with feature fields of spherical $S^2$ signals instead of SO(3)-signals. In that work it is shown that such models are computationally and memory-wise more efficient than full group convolutions over $\mathcal{X} \times \mathrm{SO}(3)$, whilst maintaining expressivity and the universal approximation property of equivariant functions (Bekkers et al., 2024, Corrolary 1.1), despite the convolution kernels—which are functions over $\mathbb{R}^3 \times S^2$—having a symmetry constraint given by $\forall_{R \in \mathrm{SO}_z(2)} : k(\mathbf{R}\mathbf{x}, \mathbf{R}\mathbf{n}) = k(\mathbf{x}, \mathbf{n})$, with $\mathrm{SO}_z(2)$ the group of rotations around the z-axis and $\mathbf{n} \in S^2$. I.e., the kernels are axially symmetric.

When factorizing such kernels into a spatial, orientation, and channel component via $k(\mathbf{x}, \mathbf{n}) := k^{(\mathbb{R}^3)}(\mathbf{x})k^{(S^2)}(\mathbf{n})k^{(channel)}$, the group convolution is split into three steps

$$L = L^{(channel)} \circ L^{(S^2)} \circ L^{(\mathbb{R}^3)} \tag{7}$$

with

$$[L^{(\mathbb{R}^3)}f](\mathbf{x}_i, \mathbf{n}) := \sum_{j \in \mathcal{N}(i)} k^{(\mathbb{R}^3)}(\mathbf{R}_\mathbf{n}^T(\mathbf{x}_j - \mathbf{x}_i)) \odot f(\mathbf{x}_i), \tag{8}$$

which is just a spatial convolution in which the kernel is a function $\mathbb{R}^3 \to \mathbb{R}^C$ that for each translation, relative to the orientation $\mathbf{n} \in S^2$, returns a $C$-dimensional vector that is element-wise multiplied with the feature at the neighboring location. Then, $L^{(S^2)}$ is a point-wise spherical convolution

$$[L^{(S^2)}f](\mathbf{x}_i) = \int_{S^2} k^{(S^2)}(\mathbf{n}^T\tilde{\mathbf{n}}) \odot f(\mathbf{x}_i, \tilde{\mathbf{n}})d\tilde{\mathbf{n}}, \tag{9}$$

again, with no channel-mixing taking place. Note that when the sphere $S^2$ is discretized with $O$ number of orientations, Eq. 9 is implemented as a point-wise matrix multiplication with a precomputed kernel matrix of size $O \times O \times C$, e.g. via `einsum("noc,poc->npc", features, spherical_kernel)`, with tensors of $f$ being of dimensions $N \times O \times C$. Finally, $L^{(channel)}$ is simply a point-wise linear layer that mixes the channels without any spatial or orientation mixing, i.e., it is distributed over all points ($N$ axis) and orientations ($O$ axis) and thus is efficiently computed.

**Handling vector input and outputs** The result of the $\mathbb{R}^3 \times S^2$-convolutional network is a spherical signal per node, and this allows to predict displacement vectors per node by simply taking a weighted sum of the spherical grid points over which the signal is sampled. I.e.,

$$\mathbf{v}_i^{out} = \sum_{\tilde{\mathbf{n}} \in \mathcal{S}} f(\mathbf{x}_i, \tilde{\mathbf{n}})\tilde{\mathbf{n}}.$$

Similarly, input vectors $\mathbf{v}_i^{in}$ at nodes $i$ can be embedded as spherical signals $f(\mathbf{x}_i, \mathbf{n}) = \mathbf{n}^T\mathbf{v}_i^{in}$ simply by taking the inner product between the reference vector (grid point) $\mathbf{n}$ and the input vector. The $\mathbb{R}^3 \times S^2$ group convolution paradigm thus allows us to build universal equivariant approximators that can predict vectors in an equivariant manner, as well as take them as inputs.

## 3 EXPERIMENTAL DESIGN

In this section, we present our hypotheses aimed at understanding the effect of equivariance, followed by architectural designs specifically crafted so that these models can evaluate the hypotheses.

### 3.1 HYPOTHESES ON THE IMPACT OF EQUIVARIANCE

Our main hypothesis is that equivariance is useful for two primary reasons:

1. Equivariance promotes **weight sharing** and is thus **data efficient**.
2. Equivariance guarantees **generalization over symmetries**.

However, we also recognize that standard (non-equivariant) neural networks may not need to rely on these properties when a large amount of data is available. In such cases, the first argument no longer applies, and the second argument becomes less significant because a large training set might be representative of the test set—or, in a sense, even include it.[1] Moreover, data augmentation might

---

[1] A highly unlikely scenario when dealing with real-world data.

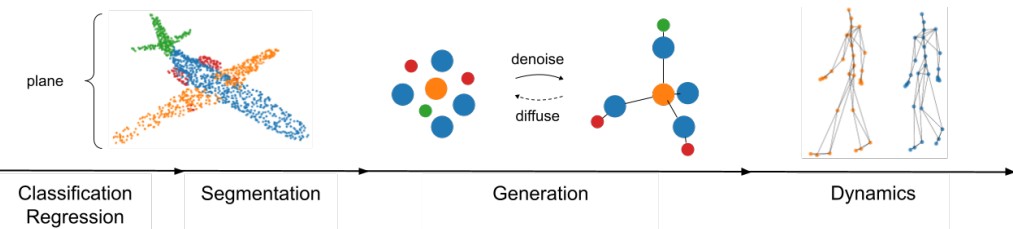

Figure 1: Progression of tasks based on geometric complexities.

be used to promote generalization over symmetries[2]. We thus formulate the following hypothesis, which will be tested with scaling experiments, with data augmentation enabled in both cases.

**Hypothesis 1 (Scaling laws)** *For tasks that require invariance or equivariance, both equivariant and non-equivariant models will converge to similar performance with increasing dataset size.*

> *(a) In the large-scale regime both equivariant and non-equivariant will perform equally well.*

> *(b) In small-scale data regimes equivariant models will outperform non-equivariant models.*

We must also consider task complexity, shown in Figure 1. We will focus the study on tasks that require either invariance or equivariance and formulate the following hypothesis to study the role of equivariance in tasks with varying geometric complexity.

**Hypothesis 2 (Geometric complexity)** *The advantages of equivariant methods over non-equivariant ones become more pronounced in tasks that require a stricter form of equivariance.*

> *(a) For invariant tasks the gap in performance between equivariant and non-equivariant models will be less pronounced than in equivariant tasks.*

> *(b) There is a hierarchy of tasks in terms of how geometrically demanding they are, and thus to what extent equivariance is needed.*

> > • ***Low complexity***: *Invariant tasks like classification and regression;*
> > • ***Moderate complexity***: *Equivariant tasks that make point-wise scalar predictions relative to a geometry (shape), like segmentation;*
> > • ***High complexity***: *Equivariant tasks that make point-wise vector predictions relative to a geometry, like denoising diffusion models and dynamics forecasting methods.*

> *The performance gap will increase with geometric complexity.*

Next, we want to investigate the effect of expanding the domain over which features are organized. In group convolutions, one typically adds an additional axis to store features that have a meaning relative to a set or grid of reference poses. For example, consider a classical CNN with a 256-dimensional feature vector per node (Table 1: rows 1-4, $16 - 19$). A G-CNN with the same number of *independent features* per node effectively has $O \times 256$ dimensional feature vectors per node and arguably more representation capacity (rows 5-11). To test the following hypothesis, models with equal representation capacity need to be compared.

**Hypothesis 3 (Representation capacity)** *Under the same representation capacity (e.g. same number of total channels per node), equivariant and non-equivariant models perform on par.*

The previous hypothesis is likely to be rejected because even when the dimensionality of the representation spaces might be the same (e.g. models 1-4 in comparison to 16-19), the invariant models are constrained to only produce invariant feature vectors and thus have a smaller *effective dimensionality*. For all models, we have universal approximation results (invariant or equivariant), so as long

---

[2]Moskalev et al. (2023) however empirically showed that at best, an apparent equivariance property is attained, which breaks under distribution shifts.

as we reach the limit of performance per model we cannot say that one model was structurally limited to finding the solution, especially when considering strictly invariant/equivariant tasks. Then, what could cause a difference in performance, if both types of models could represent the optimal solution, in principle? We thus formulate the following two hypotheses.

**Hypothesis 4 (Kernel constraint)** *We expect that unconstrained (translational) models (Eq. 3) outperform (roto-translation) models (Eq. 4) over $\mathbb{R}^3$ as the former is less constrained. Similarly, we expect that $\mathrm{SE}(3)$ equivariant group convolution methods (Eq. 7) outperform the constrained $\mathbb{R}^3$ convolutional models (Eq. 4), even when matched in representation capacity.*

Considering the above hypothesis, it might also be beneficial to break equivariance, i.e., allow a model to learn non-equivariant solutions even though it is primed for equivariant solutions. In Section 4 we test several options for breaking equivariance by providing features defined in the global coordinate system as input features. Not only will this break equivariance it will also provide the models with an *explicit representation of geometry*, whereas strictly equivariant models only have access to geometry implicitly via pair-wise relations that condition the convolution kernels. We thus formulate the following.

**Hypothesis 5 (Symmetry breaking and explicit geometric representations)** *In non-equivariant tasks, where symmetry breaking is allowed, models that are provided with explicit geometric information (such as taking coordinates as features) outperform those that do not have this information.*

### 3.2 ARCHITECTURE

Based on (Bekkers et al., 2024), we design a model class called `Rapidash` that is capable of incorporating various forms of equivariances, and thus has the flexibility to test the above hypotheses.

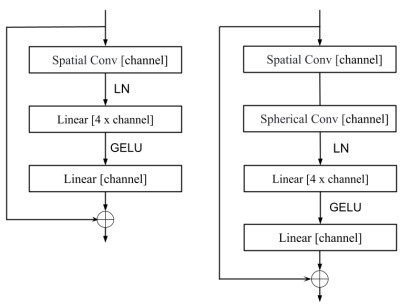

**ConvNext blocks** Our architecture is fully convolutional and is based on the ConvNext block from Liu et al. (2022). See 2 for an illustration. The block consists of the following sequence of operations: a depth-wise separable convolution, followed by a LayerNorm, then a point-wise linear layer that increases the channel dimension fourfold, a GELU activation, and a linear back to the original channel dimensions. This output is added via a skip con-

Figure 2: Block design for `Rapidash` with base space $\mathbb{R}^3$ (L) and $\mathbb{R}^3 \times S^2$ (R).

nection. The depth-wise separable convolution is either implemented via Eq. 3 or 4 for the classical (base space $\mathbb{R}^3$) architecture, and via Eq. 8 and 9 in the $\mathbb{R}^3 \times S^2$ case. Thus, the only difference to the point-cloud implementation of ConvNext is that in our group convolutional implementation, the convolution is over $\mathbb{R}^3 \times S^2$ instead of just over $\mathbb{R}^3$.

**Rapidash** To handle large point clouds, such as in the ShapeNet experiments, we need down- and up-sampling layers. Downsampling happens via strided convolution, by subsampling the point cloud using farthest point sampling, and only evaluating the convolution at the down-sampled points. Up-sampling does the inverse; it is also just a convolution layer, but sampled on a denser grid. See figure 2. The resulting model is in essence a multi-scale version of PΘNITA (Bekkers et al., 2024) and will be referred to as `Rapidash`.

## 4 EXPERIMENTS

We present a thorough analysis of the hypotheses mentioned in Section 3 through a series of experiments on various point cloud datasets like Shapenet 3D, QM9, and the CMU human motion prediction dataset, which are tasks with increasing levels of complexity (Fig. 1). For each of the experiments, we have different variations of `Rapidash` arising from equations Eq. 3 or 4 for $\mathbb{R}^3$ and Eq. 8 and 9 in $\mathbb{R}^3 \times S^2$ case, all with different input variations. For each table, the top section shows models with $\mathrm{SE}(3)$ equivariance, and the bottom section shows models with $T_3$ equivariance. Symmetry breaking is indicated by: ! ($\mathrm{SE}(3)$), ! ($T_3$ or $\mathrm{SO}(3)$), and ! (conditional $\mathrm{SO}(3)$). A

✓indicates an option is used, ✗, indicates it is not used, and "-" indicates the option is not available. ➡ marks the model with maximal equivariance and information access. All experiments use `Rapidash` under different equivariance constraints. Note, all models have implicit access to positional information via pair-wise geometric attributes. For implementation details see App. A.5. Our code is made available at *[see anonymous supplementary .zip file]*.

## 4.1 3D POINT CLOUD SEGMENTATION AND GENERATION EXPERIMENTS

In this experiment, we evaluate our models on the ShapeNet 3D dataset (Chang et al., 2015) part segmentation and generation tasks. ShapeNet consists of 16,881 shapes from 16 categories. Each shape is annotated with up to six parts, totaling 50 parts. We use the point sampling of 2,048 points and the train/validation/test split from (Qi et al., 2017). We compare our models to various *state-of-the-art* methods like PointnXt (Qian et al., 2022), Deltaconv (Wiersma et al., 2022), and GeomGCNN (Srivastava & Sharma, 2021)for the part segmentation task. For the generation task, we compare to the latent diffusion model, LION, (Zeng et al., 2022).

Table 1: Ablation study on ShapeNet 3D dataset for part segmentation task (using mean intersection over union (IOU) as the performance metric) for both the aligned and randomly rotated dataset, and shape generation experiments using the one-nearest-neighbor accuracy (1-NNA) metric. Best result per section in **blue**.

| Model Variation | Type | Coordinates as Scalars (!) | Coordinates as Vectors (!) | Normals as Scalars (!) | Normals as Vectors (!) | Global Frame (!) | Effective Equivariance | IOU↑ (aligned) | IOU↑ (rotated) | 1-NNA↓ CD | Normalized Epoch Time |
|---|---|---|---|---|---|---|---|---|---|---|---|
| **Rapidash with internal SE(3) Equivariance Constraint** | | | | | | | | | | | |
| 1 | $\mathbb{R}^3$ | ✗ | - | ✗ | - | - | SE(3) | $80.26_{\pm0.06}$ / $80.31_{\pm0.15}$ | $\mathbf{80.33_{\pm0.13}}$ / $80.20_{\pm0.07}$ | - | $\sim 1$ / $\sim 10$ |
| 2 | $\mathbb{R}^3$ | ✗ | - | ✓ | - | - | $T_3$ | $83.95_{\pm0.06}$ / $83.87_{\pm0.09}$ | $52.09_{\pm0.87}$ / $49.63_{\pm0.87}$ | - | $\sim 1$ / $\sim 10$ |
| 3 | $\mathbb{R}^3$ | ✓ | - | ✗ | - | - | none | $84.23_{\pm0.08}$ / $84.01_{\pm0.06}$ | $34.15_{\pm0.05}$ / $32.22_{\pm0.27}$ | $65.92_{\pm0.98}$ | $\sim 1$ / $\sim 10$ |
| 4 | $\mathbb{R}^3$ | ✓ | - | ✓ | - | - | none | $\mathbf{84.75_{\pm0.02}}$ / $84.48_{\pm0.16}$ | $34.07_{\pm0.43}$ / $32.90_{\pm0.47}$ | - | $\sim 1$ / $\sim 10$ |
| 5 | $\mathbb{R}^3 \times S^2$ | ✗ | ✗ | ✗ | ✗ | ✗ | SE(3) | $84.10_{\pm0.13}$ | $84.75_{\pm0.10}$ | - | $\sim 5$ |
| 6 | $\mathbb{R}^3 \times S^2$ | ✗ | ✗ | ✗ | ✗ | ✓ | SE(3) | $85.52_{\pm0.13}$ | $85.79_{\pm0.10}$ | $64.25_{\pm0.30}$ | $\sim 5$ |
| 7 | $\mathbb{R}^3 \times S^2$ | ✗ | ✗ | ✗ | ✓ | ✗ | SE(3) | $84.35_{\pm0.12}$ | $84.74_{\pm0.05}$ | - | $\sim 5$ |
| ➡ 8 | $\mathbb{R}^3 \times S^2$ | ✗ | ✗ | ✗ | ✓ | ✓ | SE(3) | $\mathbf{85.69_{\pm0.10}}$ | $\mathbf{85.81_{\pm0.06}}$ | - | $\sim 5$ |
| 9 | $\mathbb{R}^3 \times S^2$ | ✗ | ✓ | ✗ | ✗ | ✗ | SO(3) | $84.17_{\pm0.12}$ | $84.55_{\pm0.17}$ | - | $\sim 5$ |
| 10 | $\mathbb{R}^3 \times S^2$ | ✗ | ✓ | ✗ | ✗ | ✓ | SO(3) | $85.28_{\pm0.09}$ | $85.61_{\pm0.03}$ | $\mathbf{62.09_{\pm1.04}}$ | $\sim 5$ |
| 11 | $\mathbb{R}^3 \times S^2$ | ✗ | ✓ | ✗ | ✓ | ✗ | SO(3) | $84.44_{\pm0.16}$ | $84.68_{\pm0.07}$ | - | $\sim 5$ |
| 12 | $\mathbb{R}^3 \times S^2$ | ✗ | ✓ | ✗ | ✓ | ✓ | SO(3) | $85.48_{\pm0.07}$ | $85.80_{\pm0.08}$ | - | $\sim 5$ |
| 13 | $\mathbb{R}^3 \times S^2$ | ✗ | ✗ | ✓ | ✗ | ✗ | $T_3$ | $85.62_{\pm0.06}$ | $45.04_{\pm1.62}$ | - | $\sim 5$ |
| 14 | $\mathbb{R}^3 \times S^2$ | ✓ | ✗ | ✗ | ✗ | ✗ | none | $85.82_{\pm0.11}$ | $38.97_{\pm0.83}$ | $63.69_{\pm1.17}$ | $\sim 5$ |
| 15 | $\mathbb{R}^3 \times S^2$ | ✓ | ✗ | ✓ | ✗ | ✗ | none | $\mathbf{85.69_{\pm0.18}}$ | $36.08_{\pm0.75}$ | - | $\sim 5$ |
| **Rapidash with Internal $T_3$ Equivariance Constraint** | | | | | | | | | | | |
| 16 | $\mathbb{R}^3$ | ✗ | - | ✗ | - | - | $T_3$ | $85.26_{\pm0.01}$ / $85.38_{\pm0.05}$ | $31.41_{\pm0.86}$ / $32.78_{\pm0.59}$ | $67.49_{\pm0.41}$ | $\sim 1$ / $\sim 10$ |
| 17 | $\mathbb{R}^3$ | ✗ | - | ✓ | - | - | $T_3$ | $\mathbf{85.82_{\pm0.10}}$ / $85.71_{\pm0.17}$ | $\mathbf{35.42_{\pm0.98}}$ / $32.25_{\pm0.07}$ | - | $\sim 1$ / $\sim 10$ |
| 18 | $\mathbb{R}^3$ | ✓ | - | ✗ | - | - | none | $85.51_{\pm0.06}$ / $85.52_{\pm0.09}$ | $32.79_{\pm0.74}$ / $31.11_{\pm0.15}$ | $\mathbf{65.06_{\pm1.05}}$ | $\sim 1$ / $\sim 10$ |
| 19 | $\mathbb{R}^3$ | ✓ | - | ✓ | - | - | none | $85.66_{\pm0.02}$ / $85.16_{\pm0.06}$ | $33.55_{\pm0.49}$ / $30.62_{\pm0.21}$ | - | $\sim 1$ / $\sim 10$ |
| **Reference methods from literature** | | | | | | | | | | | |
| LION | - | - | - | - | - | - | - | - | - | 51.85 | - |
| PVD | - | - | - | - | - | - | - | - | - | 58.65 | - |
| DPM | - | - | - | - | - | - | - | - | - | 62.30 | - |
| DeltaConv | - | - | - | - | - | - | - | 86.90 | - | - | - |
| PointNeXt | - | - | - | - | - | - | - | 87.00 | - | - | - |
| GeomGCNN[3] | - | - | - | - | - | - | - | 89.10 | - | - | - |

## 4.2 MOLECULAR PROPERTY PREDICTION AND MOLECULE GENERATION (DISCOVERY)

For predicting molecular properties and generating molecules, we use QM9 (Ramakrishnan R., 2014), a dataset which consists of 130k small molecules and their 3-dimensional coordinates, along with molecular properties, integer-valued atom charges, and atom coordinates. It contains up to 9 heavy atoms and 29 atoms including hydrogens. We use the train/val/test partitions introduced in Gilmer et al. (2017), which consists of 100K/18K/13K samples respectively for each partition. We evaluate the prediction of molecular properties using the MAE metric and compare these with EGNN (Satorras et al., 2021), Dimenet++ (Gasteiger et al., 2022) and SE(3)- Transformer Fuchs et al. (2020)

---

[3]GeomGCNN is trained with 1024 points instead of 2048.

Table 2: Ablation study on QM9 for property prediction task using mean absolute error (MAE) as the performance metric for three properties *mu*, *alpha* and $\epsilon_{HOMO}$, as well as molecule generation task (discovery) using atom stability, molecule stability and *discover* as performance metric.

| Model Variation | Type | Coordinates as Scalars [!] | Coordinates as Vectors [!] | Effective Equivariance | MAE $\mu$ (D) | MAE $\alpha$ ($a_0^3$) | MAE $\epsilon_{HOMO}$ (meV) | Stab Atom % | Stab Mol % | Discover % | Normalized Epoch Time |
|---|---|---|---|---|---|---|---|---|---|---|---|
| | | | | | | | | | | | |
| | | | | *Rapidash* with internal SE(3) Equivariance Constraint | | | | | | | |
| | | | | | **Regression** | | | **Generation** | | | |
| 1 | $\mathbb{R}^3$ | ✗ | - | SE(3) | 0.0609±0.0001 / **0.0185**±0.0005 | 0.1196 | 24.21±0.20 / 31.87 | - | - | - | ~1 / ~20 |
| 2 | $\mathbb{R}^3$ | ✓ | - | none | 0.0183±0.0003 | **0.0544**±0.0001 / 0.1195 | **23.17**±0.3 / 32.21 | 97.50±0.68 | 77.01±1.14 | **90.18**±0.87 | ~1 / ~20 |
| 3 | $\mathbb{R}^3 \times S^2$ | ✗ | ✗ | SE(3) | 0.0109±0.0005 | 0.034±0.10 | 20.43±.59 | 99.02±0.09 | 91.12±.41 | **91.61**±.188 | ~10 |
| → 4 | $\mathbb{R}^3 \times S^2$ | ✗ | ✓ | SO(3) | 0.0107±0.001 | 0.0397±0.0011 | 18.80±3.24 | **99.29**±.02 | **92.34**±.20 | 91.13±.35 | ~10 |
| 5 | $\mathbb{R}^3 \times S^2$ | ✓ | ✗ | none | **0.009**±0.0007 | **0.038**±0.005 | **18.42**±0.31 | 97.77±.02 | 91.11±.53 | 90.42±.84 | ~10 |
| | | | | *Rapidash* with Internal $T_3$ Equivariance Constraint | | | | | | | |
| 6 | $\mathbb{R}^3$ | ✗ | - | $T_3$ | **0.0615**±0.005 / 0.02196±0.023 | 0.0940 | 28.02±.78 / 38.15 | 98.90±.03 | 86.1±.61 | **92.89**±.32 | ~1 / ~19 |
| 7 | $\mathbb{R}^3$ | ✓ | - | none | **0.0219**±0.0002 / 0.0672±0.001 | 0.2051 | **27.54**±.68 / 38.63 | **98.91**±.01 | **85.99**±.38 | 90.05±.50 | ~1 / ~19 |
| | | | | Reference methods from literature | | | | | | | |
| EGNN | - | - | - | - | 0.0290 | 0.0710 | 29.00 | 98.7 | 82.0 | - | - |
| DimeNet++* | - | - | - | - | 0.0297 | 0.0435 | 24.60 | - | - | - | - |
| SE(3)-T | - | - | - | - | 0.1420 | 0.0510 | 53.00 | - | - | - | - |
| MiDi (adaptive)** | - | - | - | - | - | - | - | 99.8 | 97.5 | 64.5 | - |
| Clifford-EDM | - | - | - | - | - | - | - | 99.5 | 93.4 | 89.5 | - |
| EGNN-EDM | - | - | - | - | - | - | - | 99.25 | 90.73 | 89.5 | - |
| PΘNITA | - | - | - | - | - | - | - | 98.9 | 87.8 | - | - |

\* Not a fair comparison as trained for longer. \*\*Not a fair comparison, as MiDi uses open Babel optimization procedure.

For the molecule generation *(discovery)* task, we train an equivariant generative model that uses equivariant denoising layers in a diffusion model like that in Karras et al. (2022) to unconditionally generate molecules. We evaluate molecular generation on metrics from Hoogeboom et al. (2022) like atomic stability and molecule stability. We introduce a new metric, ***discover***, defined as a product of **validity** × **unique** × **novelty**, which represents a *discovery* rate for a generated sample. It represents the fraction of generated samples that are jointly valid, unique, and new, which is crucial for discovering new molecules. We compare our models with EGNN-based denoising diffusion (Ho et al., 2020) model like Hoogeboom et al. (2022), , MiDi (Vignac et al., 2023), and PΘNITA (Bekkers et al., 2024) based diffusion model.

### 4.3 HUMAN MOTION PREDICTION TASK

In table 3, we evaluate our models on the CMU Human Motion Capture dataset (Gross & Shi, 2001), consisting of 31 equally connected nodes, each representing a specific position on the human body during walking. Given node positions at a random frame, the objective is to predict node positions after 30 timesteps. As per Huang et al. (2022) we use the data of the 35th human subject for the experiment. We compare our models to NRI (Kipf et al., 2018), EGNN (Satorras et al., 2021), CEGNN (Ruhe et al., 2023) and CSMPN Liu et al. (2024) and show improved performance. We demonstrate that equivariant architectures enable better performance in motion prediction. See Fig:5 in Appendix.

## 5 RESULTS

**Hypothesis 1 is accepted**: *Equivariant methods are more data-efficient then their non-equivariant counter part*. The hypothesis is tested on Shapenet segmentation for various dataset size regimes as shown in Fig. 3 and Tab. 6 in the Appendix.

**Hypothesis 2 is accepted**: *The advantage of equivariant models gets more pronounced with task complexity*. Although, on all experiments the equivariant models have an edge over non-equivariant models, this improvement is more pronounced in complex and equivariant tasks, such as QM9 regression and generation (Tab. 2), Shapenet generation (Tab. 1) and motion prediction (Tab. 3), in which we note that both Shapenet generation and CMU motion prediction do not require equivariance, but are complex geometric tasks.

**Hypothesis 3 is rejected**: *Increasing model capacity by increasing the number of channels does not lead to a performance gain as seen by the group convolutional methods*. The performance of low vs high capacity (compared gray vs regular numbers in the tables) is often negligible. The results suggest that all of the models are already maxed-out in net capacity to reach optimal performance.

Table 3: Ablation study on Human motion prediction task on CMU Motion Capture dataset. We evaluate our models using the mean squared error ($\times 10^{-2}$) metric.

| | | `Rapidash` with internal $SE(3)$ Equivariance Constraint | | | | | | | | |
|---|---|---|---|---|---|---|---|---|---|---|
| Model Variation | Type | ! Coordinates as Scalars | Coordinates as Vectors | ! Velocity as Scalars | Velocity as Vectors | ! Global Frame | Effective Equivariance | MSE | MSE (rotated) | Normalized Epoch Time |
| 1 | $\mathbb{R}^3$ | ✗ | - | ✗ | - | - | SE(3) | $>100$ / $>100$ | $>100$ / $>100$ | $\sim 1$ / $\sim 20$ |
| 2 | $\mathbb{R}^3$ | ✗ | - | ✓ | - | - | $T_3$ | $5.44_{\pm 0.12}$ / $\mathbf{4.81}_{\pm 0.13}$ | $24.45_{\pm 0.66}$ / $\mathbf{25.66}_{\pm 1.53}$ | $\sim 1$ / $\sim 20$ |
| 3 | $\mathbb{R}^3$ | ✓ | - | ✗ | - | - | none | $6.88$ / $5.93$ | $>100$ / $84.54$ | $\sim 1$ / $\sim 20$ |
| 4 | $\mathbb{R}^3$ | ✓ | - | ✓ | - | - | none | $5.93$ / $5.53$ | $>100$ / $>100$ | $\sim 1$ / $\sim 19$ |
| 5 | $\mathbb{R}^3 \times S^2$ | ✗ | ✗ | ✗ | ✗ | ✗ | SE(3) | $6.17$ | $8.71$ | $\sim 4$ |
| 6 | $\mathbb{R}^3 \times S^2$ | ✗ | ✗ | ✗ | ✓ | ✗ | SE(3) | $\mathbf{4.62}_{\pm 0.02}$ | $\mathbf{4.70}_{\pm 0.09}$ | $\sim 4$ |
| → 7 | $\mathbb{R}^3 \times S^2$ | ✗ | ✗ | ✗ | ✓ | ✓ | SE(3) | $4.95$ | $6.71$ | $\sim 4$ |
| 8 | $\mathbb{R}^3 \times S^2$ | ✗ | ✓ | ✗ | ✗ | ✗ | SO(3) | $7.26$ | $10.03$ | $\sim 4$ |
| 9 | $\mathbb{R}^3 \times S^2$ | ✗ | ✓ | ✗ | ✗ | ✓ | SO(3) | $4.95$ | $7.57$ | $\sim 4$ |
| 10 | $\mathbb{R}^3 \times S^2$ | ✗ | ✓ | ✗ | ✓ | ✗ | SO(3) | $5.31$ | $7.99$ | $\sim 4$ |
| 11 | $\mathbb{R}^3 \times S^2$ | ✗ | ✓ | ✗ | ✓ | ✓ | SO(3) | $5.14$ | $7.50$ | $\sim 4$ |
| 12 | $\mathbb{R}^3 \times S^2$ | ✗ | ✗ | ✓ | ✗ | ✗ | $T_3$ | $\mathbf{4.77}_{\pm 0.14}$ | $\mathbf{31.34}_{\pm 0.67}$ | $\sim 4$ |
| 13 | $\mathbb{R}^3 \times S^2$ | ✓ | ✗ | ✗ | ✗ | ✗ | none | $5.29$ | $60.82$ | $\sim 4$ |
| 14 | $\mathbb{R}^3 \times S^2$ | ✓ | ✗ | ✓ | ✗ | ✗ | none | $5.31$ | $65.80$ | $\sim 4$ |
| | | `Rapidash` with Internal $T_3$ Equivariance Constraint | | | | | | | | |
| 15 | $\mathbb{R}^3$ | ✗ | - | ✗ | - | - | $T_3$ | $6.53$ / $5.99$ | $43.80$ / $>100$ | $\sim 1$ / $\sim 19$ |
| 16 | $\mathbb{R}^3$ | ✗ | - | ✓ | - | - | $T_3$ | $\mathbf{5.3}_{\pm 0.04}$ / $24.32_{\pm 1.97}$ | $\mathbf{40.69}_{\pm 0.87}$ / $>100_{\pm 6.65}$ | $\sim 1$ / $\sim 19$ |
| 17 | $\mathbb{R}^3$ | ✓ | - | ✗ | - | - | none | $6.03$ / $6.82$ | $77.78$ / $69.42$ | $\sim 1$ / $\sim 19$ |
| 18 | $\mathbb{R}^3$ | ✓ | - | ✓ | - | - | none | $5.49$ / $5.54$ | $66.47$ / $76.12$ | $\sim 1$ / $\sim 19$ |
| | | Reference methods from literature | | | | | | | | |
| TFN | - | - | - | - | - | - | - | $66.90$ | - | - |
| EGNN | - | - | - | - | - | - | - | $31.70$ | - | - |
| CGENN | - | - | - | - | - | - | - | $9.41$ | - | - |
| CSMPN | - | - | - | - | - | - | - | $7.55$ | - | - |

**Hypothesis 4 is accepted**: *For equivariant tasks, having less constraints on the model is beneficial.* See, e.g., Tab. 1, comparing models 1-4 with models 16-19, the latter being less constrained, and performing significantly better. Consider also the QM9 experiments (Tab. 2) which is an equivariant task. All models 1,2 and 5-12 are equivariant and universal approximators and should be able to solve the problem. Models 1-2 however use constrained kernels (symmetric, distance-based) while the group convolutional models are not. The latter largely outperforms the former.

**Hypothesis 5 is accepted**: *Providing explicit geometric information can improve performance, even when breaking symmetry*. This is, e.g., observed in the tables by comparing the group convolutional models that take either coordinates as scalars (breaks equivariance) or as vectors (maintains equvariance) against those models that do not take any coordinate information as input.

## 6 DISCUSSION AND CONCLUSION

**Discussion** Beyond hypothesis insights, our basic convolutional architecture (`Rapidash`) performs exceptionally well on all considered tasks, often reaching state-of-the-art (SotA) performance, s.a. on ShapeNet/QM9 generation and CMU motion prediction. Surprisingly, even non-equivariant methods perform well on equivariant tasks, though not at SotA level and without equivariant generalization guarantees. These results support the "convolution is all you need" idea (Cohen et al., 2019; Bekkers et al., 2024), contrasting the popular "attention-is-all-you-need" claim (Vaswani, 2017). The generalizability of the presented hypotheses remains to be tested outside of class of convolutional architectures, to which `Rapidash` belongs. **Conclusion** In this work, we conducted a comprehensive evaluation of equivariant and non-equivariant models across various tasks of varying geometric complexities. In the debate of equivariant vs non-equivariant models, we provide strong evidence in favor of introducing structure over mere scaling. Specifically, we find that equivariant models (1) are more data efficient, (2) increasingly outpace non-equivariant models when increasing task geometric complexity, and (3) see performance benefits from targeted symmetry breaking.

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

# A APPENDIX

## A.1 MATHEMATICAL PREREQUISITES AND NOTATIONS

**Groups.** A *group* is an algebraic structure defined by a set $G$ and a binary operator $\cdot : G \times G \to G$, known as the *group product*. This structure $(G, \cdot)$ must satisfy four axioms: (1) *closure*, where $\forall_{h,g \in G} : h \cdot g \in G$; (2) the existence of an *identity* element $e \in G$ such that $\forall_{g \in G}, e \cdot g = g \cdot e = g$, (3) the existence of an *inverse* element, i.e. $\forall_{g \in G}$ there exists a $g^{-1} \in G$ such that $g^{-1} \cdot g = e$; and (4) *associativity*, where $\forall_{g,h,p \in G} : (g \cdot h) \cdot p = g \cdot (h \cdot p)$. Going forward, group product between two elements will be denoted as $g, g' \in G$ by juxtaposition, i.e., as $g\,g'$.

For Special Euclidean group $SE(n)$, the group product between two roto-translations $g = (\mathbf{x}, \mathbf{R})$ and $g' = (\mathbf{x}', \mathbf{R}')$ is given by $(\mathbf{x}, \mathbf{R})\,(\mathbf{x}', \mathbf{R}') = (\mathbf{R}\mathbf{x}' + \mathbf{x}, \mathbf{R}\,\mathbf{R}')$, and its identity element is given by $e = (\mathbf{0}, \mathbf{I})$.

**Homogeneous Spaces.** A group can act on spaces other than itself via a *group action* $\mathcal{T} : G \times X \to X$, where $X$ is the space on which $G$ acts. For simplicity, the action of $g \in G$ on $x \in X$ is denoted as $g\,x$. Such a transformation is called a group action if it is homomorphic to $G$ and its group product. That is, it follows the group structure: $(g\,g')\,x = g\,(g'\,x)\ \forall g, g' \in G, x \in X$, and $e\,x = x$. For example, consider the space of 3D positions $X = \mathbb{R}^3$, e.g., atomic coordinates, acted upon by the group $G = SE(3)$. A position $\mathbf{p} \in \mathbb{R}^3$ is roto-translated by the action of an element $(\mathbf{x}, \mathbf{R}) \in SE(3)$ as $(\mathbf{x}, \mathbf{R})\,\mathbf{p} = \mathbf{R}\,\mathbf{p} + \mathbf{x}$.

A group action is termed *transitive* if every element $x \in X$ can be reached from an arbitrary origin $x_0 \in X$ through the action of some $g \in G$, i.e., $x = gx_0$. A space $X$ equipped with a transitive action of $G$ is called a *homogeneous space* of $G$. Finally, the *orbit* $G\,x := \{g\,x \mid g \in G\}$ of an element $x$ under the action of a group $G$ represents the set of all possible transformations of $x$ by $G$. For homogeneous spaces, $X = G\,x_0$ for any arbitrary origin $x_0 \in X$.

**Quotient spaces.** The aforementioned space of 3D positions $X = \mathbb{R}^3$ serves as a homogeneous space of $G = SE(3)$, as every element $\mathbf{p}$ can be reached by a roto-translation from $\mathbf{0}$, i.e., for every $\mathbf{p}$ there exists a $(\mathbf{x}, \mathbf{R})$ such that $\mathbf{p} = (\mathbf{x}, \mathbf{R})\,\mathbf{0} = \mathbf{R}\,\mathbf{0} + \mathbf{x} = \mathbf{x}$. Note that there are several elements in $SE(3)$ that transport the origin $\mathbf{0}$ to $\mathbf{p}$, as any action with a translation vector $\mathbf{x} = \mathbf{p}$ suffices regardless of the rotation $\mathbf{R}$. This is because any rotation $\mathbf{R}' \in SO(3)$ leaves the origin unaltered.

We denote the set of all elements in $G$ that leave an origin $x_0 \in X$ unaltered the *stabilizer subgroup* $\mathrm{Stab}_G(x_0)$. In subsequent analyses, the symbol $H$ is used to denote the stabilizer subgroup of a chosen origin $x_0$ in a homogeneous space, i.e., $H = \mathrm{Stab}_G(x_0)$. We further denote the *left coset* of $H$ in $G$ as $g\,H := \{g\,h \mid h \in H\}$. In the example of positions $\mathbf{p} \in X = \mathbb{R}^3$ we concluded that we can associate a point $\mathbf{p}$ with many group elements $g \in SE(3)$ that satisfy $\mathbf{p} = g\,\mathbf{0}$. In general, letting $g_x$ be any group element s.t. $x = g_x\,x_0$, then any group element in the left set $g_x\,H$ is also identified with the point $\mathbf{p}$. Hence, any $x \in X$ can be identified with a left coset $g_x H$ and vice versa.

Left cosets $g\,H$ then establish an *equivalence relation* $\sim$ among transformations in $G$. We say that two elements $g, g' \in G$ are equivalent, i.e., $g \sim g'$, if and only if $g\,x_0 = g'\,x_0$. That is, if they belong to the same coset $g\,H$. The space of left cosets is commonly referred to as the *quotient space* $G/H$.

We consider *feature maps* $f : X \to \mathbb{R}^C$ as multi-channel signals over homogeneous spaces $X$. Here, we treat point clouds as sparse feature maps, e.g., sampled only at atomic positions. In the general continuous setting, we denote the space of feature maps over $X$ with $\mathcal{X}$. Such feature maps undergo group transformations through *regular group representations* $\rho^{\mathcal{X}}(g) : \mathcal{X} \to \mathcal{X}$ parameterized by $g$, and which transform functions $f \in \mathcal{X}$ via $[\rho^{\mathcal{X}}(g)f](x) = f(g^{-1}x)$.

## A.2 RETHINKING EQUIVARIANT MESSAGE PASSING AS GROUP CONVOLUTION

**Message Passing** Consider the graph representation of a point cloud $\mathcal{G} = (V, E)$ with points in a homogeneous space $X$. Label nodes as $i \in V$ and edges between nodes as $(i, j) \in E$. Each node $i$ has a corresponding coordinate $x_i \in X$. To each node, a feature $f_i \in \mathbb{R}^C$ is associated, forming a discrete analog to the previously defined dense feature maps $f : X \to \mathbb{R}^C$, where $f_i = f(x_i)$. Thus, the features associated with a node $i$ are $(x_i, f_i)$. Additionally, one can associate attributes to the edges between nodes as $a_{ij}$.

Message passing is defined in three steps as follows:

$$(1) \text{ Compute messages} \quad m_{ij} = \phi_m(f_i, f_j, a_{ij})$$

$$(2) \text{ Aggregate messages} \quad m_i = \sum_{j \in \mathcal{N}(i)} m_{ij}$$

$$(3) \text{ Update features} \quad f_i^{out} = \phi_u(f_i, m_i)$$

where $\mathcal{N}(i) = \{j \mid (i, j) \in E\}$ denotes the set of nodes in the neighborhood of the $i^{th}$ node.

**Group Convolution as Equivariant Message Passing.**

Group convolution can be written in terms of the message passing formalism. Given the graph $\mathcal{G} = (V, E)$ defined above, group convolution can be written as the sum

$$[\phi(\mathcal{G})](g_{x_i}) = \sum_{j \in \mathcal{N}(i)} k(g_{x_i}^{-1} x_j) f_j$$

Defining $a_{ij} = g_{x_i}^{-1} x_j$ as the edge feature between a pair of nodes, (1) the message function is $\phi_m(f_i, f_j, a_{ij}) = k(g_{x_i}^{-1} x_j) f_j$, (2) the permutation invariant aggregation function is the sum, and (3) the update map is $\phi_u(f_i, m_i) = m_i$. Indeed, the message function is a linear map which is determined by the attribute $a_{ij}$, in this case, determined by the relative pose between two nodes. This dependence on only relative pose underlies both the equivariance and weight sharing of equivariant message passing.

**Equivalence Classes and Invariant Attributes.** Equivalence classes of position-orientation coordinates with $[x_i, x_j]$ is defined as

$$[x_i, x_j] = \{(x_i', x_j') \in X \times X \mid (x_i', x_j') \sim (x_i, x_j)\}$$

where equivalence classes are given by equivalence under group action on each element, i.e.

$$(x_i, x_j) \sim (x_i', x_j') \iff \exists g \in G : (x_i', x_j') = (gx_i, gx_j)$$

The list of invariant attributes for position orientation that determine the equivalence relations are given below for completeness (Bekkers et al., 2024).

$$
\begin{aligned}
\mathbb{R}^2 \text{ and } \mathbb{R}^3 : & \quad [\mathbf{p}_i, \mathbf{p}_j] & \mapsto & \quad a_{ij} = \|\mathbf{p}_j - \mathbf{p}_i\|, \\
\mathbb{R}^2 \times S^1 \text{ and } SE(2) : & \quad [(\mathbf{p}_i, \mathbf{o}_i), (\mathbf{p}_j, \mathbf{o}_j)] & \mapsto & \quad a_{ij} = \left(\mathbf{R}_{\mathbf{o}_i}^{-1}(\mathbf{p}_j - \mathbf{p}_i), \arccos \mathbf{o}_i^\top \mathbf{o}_j\right), \\
\mathbb{R}^3 \times S^2 : & \quad [(\mathbf{p}_i, \mathbf{o}_i), (\mathbf{p}_j, \mathbf{o}_j)] & \mapsto & \quad a_{ij} = \begin{pmatrix} \mathbf{o}_i^\top(\mathbf{p}_j - \mathbf{p}_i) \\ \|(\mathbf{p}_j - \mathbf{p}_i) - \mathbf{o}_i^\top(\mathbf{p}_j - \mathbf{p}_i)\mathbf{o}_i\| \\ \arccos \mathbf{o}_i^\top \mathbf{o}_j \end{pmatrix}, \\
SE(3) : & \quad [(\mathbf{p}_i, \mathbf{R}_i), (\mathbf{p}_j, \mathbf{R}_j)] & \mapsto & \quad a_{ij} = \left(\mathbf{R}_i^{-1}(\mathbf{p}_j - \mathbf{p}_i), \mathbf{R}_i^{-1}\mathbf{R}_j\right).
\end{aligned}
$$

## A.3 Internal vs External Symmetry Breaking

In this section, we provide an analysis of symmetry breaking in equivariant neural networks, distinguishing between what we consider to be two fundamentally different types: *external* and *internal* symmetry breaking.

### A.3.1 External Symmetry Breaking

External symmetry breaking occurs when an inherently equivariant architecture loses its equivariance properties due to the way inputs are provided to the network. Consider a linear layer $L$ designed to be $G$-equivariant for some group $G$ (e.g., SE(3)). Let $v$ be a vector that transforms under the group action, and define $x_g = g \cdot v$ as its transformed version in global coordinates. When these

coordinates are provided as scalar triplets, they are processed independently with no relation to the group action, thus:

$$x \neq x_g \quad \implies \quad L(x_g) \neq L(x) \tag{10}$$

This means the network processes transformed inputs differently, breaking equivariance. In contrast, when inputs are properly specified as vectors that transform under the group action, we maintain equivariance:

$$L(g \cdot v) = g \cdot L(v) \quad \forall g \in G \tag{11}$$

Moreover, for truly invariant features (such as one-hot encodings of atom types in QM9), the group action is trivial:

$$g \cdot x = x \quad \implies \quad L(g \cdot x) = L(x) \tag{12}$$

which guarantees invariance of the entire network to group transformations.

### A.3.2 INTERNAL SYMMETRY BREAKING

Internal symmetry breaking then refers to the deliberate relaxation of equivariance constraints within the layers themselves. Recent works have explored various approaches to this, including:

- Basis decomposition methods that mix equivariant and non-equivariant components
- Learnable deviation from strict equivariance through regularization
- Progressive relaxation of equivariance constraints during training

In some cases, an internally symmetry-broken layer $L_{\text{broken}}$ can be expressed as:

$$L_{\text{broken}} = L_{\text{equiv}} + \alpha L_{\text{non-equiv}} \tag{13}$$

where $\alpha$ controls the degree of symmetry breaking, some approaches (e.g., (Wang et al., 2022)) implement schemes where $\alpha$ is annealed during training to gradually enforce stricter equivariance.

### A.3.3 COMBINING INTERNAL AND EXTERNAL BREAKING

In practice, both types of symmetry breaking often occur simultaneously and can interact. For example, a network may use non-stationary filters (internal breaking) while also accepting coordinate inputs in a global reference frame (external breaking). The total degree of symmetry breaking then depends on both mechanisms. Theoretical quantitative bounds on models with task-specific symmetries show that aligning data symmetry with architecture leads to the best model, although having partial or approximate equivariance can improve generalization (Petrache & Trivedi, 2023).

### A.3.4 RELATIONSHIP TO OUR WORK

While our study primarily focuses on external symmetry breaking (cf. the red exclamation marks in our tables), we note that the transition between translation-equivariant layers (Table 1, rows 15-18) and roto-translation equivariant layers (rows 1-4) could be viewed as a form of internal symmetry breaking. However, we distinguish our approach from explicit internal symmetry-breaking methods as we do not implement continuous relaxation of equivariance constraints within layers, but rather compare discrete architectural choices with different symmetry properties.

The experiments demonstrate that carefully controlled symmetry breaking, whether internal or external, can significantly improve model performance when the underlying data exhibits only approximate symmetries. This aligns with recent findings in the field showing the benefits of relaxed equivariance constraints (van der Ouderaa et al., 2022; Kim et al., 2023; Pertigkiozoglou et al., 2024).

### A.4 ADDITIONAL RESULTS AND DISCUSSION

**Modelnet40C dataset** We present experiments for 9 different model variations for ModelNet40-C Sun et al. (2022), a dataset designed to evaluate robustness in point-cloud methods using different levels of corruptions to the ModelNet dataset. It mainly consists of three different types of corruptions: density corruptions, Noise corruptions and Transformation corruptions. We present Modelnet40 classification results in Tab. 4 and Modelnet40C classification results in Tab. 5.

**Practical implications of the results** Our experiments on ModelNet40 demonstrate similar trends to those discussed in the main paper for the experiments on ShapeNet, QM9 and CMU motion datasets. For a non-equivariant task (low geometric complexity) like classification, we see that equivariant methods perform best, model 6-7 in Tab . 4. For ModelNet40C, which presents different corruptions, we see that the error rates for rotation transformation corruptions are lower than the other corruptions. Additionally, we see that translation equivariant methods models 8-9, in Tab. 5 perform best across various corruptions (except rotation).

Table 4: Ablation study on ModelNet40 for classification task using accuracy as metric.

| Model Variation | Type | Coordinates as Scalars ! | Coordinates as Vectors ! | Global Frame ! | Effective Equivariance | Accuracy (%) |
|---|---|---|---|---|---|---|
| \multicolumn{7}{c}{Rapidash with internal SE(3) Equivariance Constraint} | | | | | | |
| 1 | $\mathbb{R}^3$ | ✗ | - | - | SE(3) | 73.41 |
| 2 | $\mathbb{R}^3$ | ✓ | - | - | none | 81.84 |
| 3 | $\mathbb{R}^3 \times S^2$ | ✗ | ✗ | ✗ | SE(3) | - |
| 4 | $\mathbb{R}^3 \times S^2$ | ✗ | ✗ | ✓ | SO(3) | 82.13 |
| 5 | $\mathbb{R}^3 \times S^2$ | ✗ | ✓ | ✗ | none | 84.00 |
| 6 | $\mathbb{R}^3 \times S^2$ | ✗ | ✓ | ✓ | none | 84.84 |
| 7 | $\mathbb{R}^3 \times S^2$ | ✓ | ✗ | ✗ | none | 82.73 |
| \multicolumn{7}{c}{Rapidash with Internal $T_3$ Equivariance Constraint} | | | | | | |
| 8 | $\mathbb{R}^3$ | ✗ | - | - | $T_3$ | 83.87 |
| 9 | $\mathbb{R}^3$ | ✓ | - | - | none | 83.00 |
| \multicolumn{7}{c}{Reference methods from literature} | | | | | | |
| PointNet* | - | - | - | - | - | 90.7 |
| PointNet++* | - | - | - | - | - | 93.0 |
| DGCNN* | - | - | - | - | - | 92.6 |

\* Not a fair comparison as trained for much longer.

**Experiments with data reduction in training** Experiments in data reduction show that the model variant with greatest equivariance, and the most non-symmetry-breaking information (SE3-pV-100%) performed the best, and suffered from the least performance degradation when trained on a smaller percentage of the training set. See Figure 3 for training curves.

### A.5 IMPLEMENTATION DETAILS

We implemented our models using PyTorch (Paszke et al., 2019), utilizing PyTorch-Geometric's message passing and graph operations modules (Fey & Lenssen, 2019), and employed Weights and Biases for experiment tracking and logging. A pool of GPUs, including A100, A6000, A5000, and 1080Ti, was utilized as computational units. To ensure consistent performance across experiments, computation times were carefully calibrated, maintaining GPU homogeneity throughout.

For all experiments, we use Rapidash with 7 layers with 0 fiber dimensions for $\mathbb{R}^3$ and 0 or 8 fiber dimensions for $\mathbb{R}^3 \times S^2$. The polynomial degree was set to 2. We used the Adam optimizer (Kingma & Ba, 2014), with a learning rate of $1e-4$, and with a CosineAnealing learning rate schedule with a warmup period of 20 epochs.

Table 5: Model performance (% error rate) under different types of corruptions for ModeNet40-C dataset. **MV** corresponds to the model variation listed out in Tab. 4

| Density Corruptions | | | | | |
|---|---|---|---|---|---|
| MV | Occlusion | LiDAR | Density Inc. | Density Dec. | Cutout |
| 1 | 80.3 | 87.5 | - | 68.0 | 57.4 |
| 2 | 62.0 | 69.3 | - | 30.8 | 26.1 |
| 3 | 71.97 | 79.54 | | 33.0 | 29.1 |
| 4 | 67.5 | 76.2 | - | 27.9 | 27.0 |
| 5 | 67.4 | 74.3 | - | 30.0 | 26.7 |
| 6 | 65.4 | 73.9 | - | 25.3 | 24.3 |
| 7 | 59.6 | 66.3 | - | 27.2 | 25.0 |
| 8 | 57.8 | 56.0 | - | 24.2 | 22.1 |
| 9 | 53.1 | 52.9 | - | 22.5 | 20.6 |

| Noise Corruptions | | | | | | Transformation Corruptions | | | | | |
|---|---|---|---|---|---|---|---|---|---|---|---|
| MV | Uniform | Gauss. | Impulse | Upsamp. | Backgr. | MV | Rotation | Shear | FFD | RBF | Inv. RBF |
| 1 | 46.3 | 55.1 | 82.4 | 41.2 | 96.2 | 1 | 37.1 | 41.0 | 42.4 | - | - |
| 2 | 20.2 | 20.9 | 41.5 | 20.0 | 95.9 | 2 | 24.7 | 22.0 | 22.9 | - | - |
| 3 | 24.0 | 26.5 | 39.3 | 22.4 | 94.8 | 3 | 18.6 | 22.6 | 24.1 | - | - |
| 4 | 20.2 | 21.3 | 38.3 | 20.5 | 95.7 | 4 | 18.8 | 23.2 | 24.0 | - | - |
| 5 | 20.9 | 23.2 | 45.6 | 20.3 | 94.5 | 5 | 18.2 | 23.1 | 22.9 | - | - |
| 6 | 20.9 | 22.9 | 44.4 | 20.7 | 95.9 | 6 | 17.2 | 21.5 | 21.5 | - | - |
| 7 | 19.3 | 20.7 | 32.0 | 19.7 | 95.5 | 7 | 24.7 | 22.5 | 23.0 | - | - |
| 8 | 18.8 | 20.5 | 39.5 | 18.9 | 95.3 | 8 | 22.5 | 19.4 | 20.9 | - | - |
| 9 | 18.7 | 20.1 | 32.1 | 19.1 | 92.7 | 9 | 23.2 | 20.6 | 20.9 | - | - |

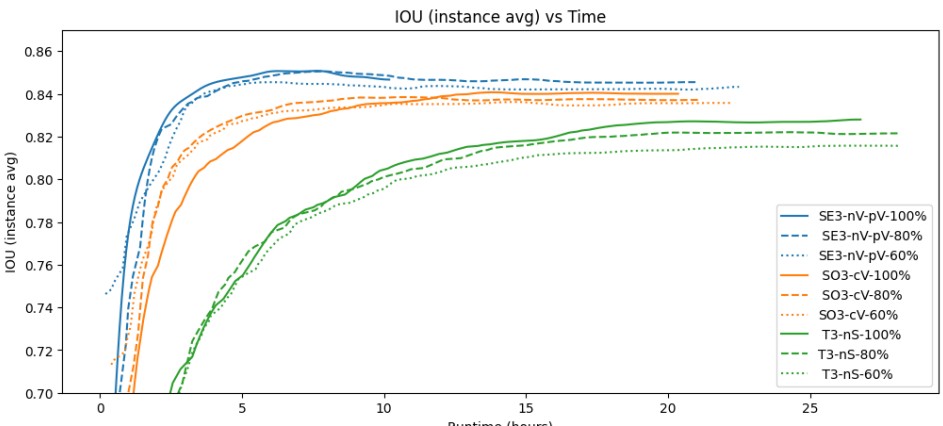

Figure 3: Smoothed IOU (Instance average) performance curves over time for ShapeNet part segmentation. The legend codes correspond to the effective equivariance, types of features being passed in (c = coordinates, n = normals, p = poses, S = as scalars, V = as vectors), and percent of the training dataset being trained on.

**ShapeNet 3D (Generation).** For this task, we trained we trained model variation (1-4 & 16-19) in Tab.2 with two different settings of hidden features C = 256 (gray) and C = 2048. The later inflated model was trained to match the representation capacity of the rest of the models. For segmentation task, we use rotated samples and compute IOU with aligned and rotated samples. All the models were trained for 500 epochs with a learning rate $5e - 3$ and weight decay of $1e - 8$.

**QM9 (Segmentation and Generation).** For the segmentation task, we trained we trained model variation (1,2 & 6,7) in Tab.2 with two different settings of hidden features C = 256 (gray) and C = 2048. The later inflated model was trained to match the representation capacity of the rest of the models. All the models were trained for 500 epochs with a learning rate $5e - 3$ and weight decay of $1e - 8$.

**CMU Motion Prediction.** For this task we trained model variation (1-4 & 16-19) in Tab.3 with two different settings of hidden features C = 256 (gray) and C = 2048. The later inflated model was

Table 6: Ablation study on ShapeNet 3D part segmentation using instance mean IOU as the performance metric for randomly rotated dataset, comparing validation-set performance when trained on different percentages of the training dataset

| | | Rapidash with internal SE(3) Equivariance Constraint | | | | | | | | | Normalized |
|---|---|---|---|---|---|---|---|---|---|---|---|
| Model Variation | Type | Coordinates as Scalars | Coordinates as Vectors | Normals as Scalars | Normals as Vectors | Global Frame | Effective Equivariance | IOU↑ 100% | IOU↑ 80% | IOU↑ 60% | Epoch Time |
| ➡ 8 | $\mathbb{R}^3 \times S^2$ | ✗ | ✗ | ✗ | ✓ | ✓ | SE(3) | 85.45 | 85.46 | 85.10 | |
| 9 | $\mathbb{R}^3 \times S^2$ | ✗ | ✓ | ✗ | ✗ | ✗ | SO(3) | 84.48 | 84.21 | 84.07 | - |
| | | Rapidash with Internal $T_3$ Equivariance Constraint | | | | | | | | | |
| 17 | $\mathbb{R}^3$ | ✗ | - | ✓ | - | - | $T_3$ | 83.00 | 82.50 | 82.03 | - |

trained to match the representation capacity of the models (5-15). All the models were trained for 1000 epochs with a learning rate $5e-3$ and weight decay of $1e-8$.

**ModelNet40 and ModelNet40C (Classifcation)** For this task we trained 9 model variations presented in Tab.4 and Tab. 5. All the models were trained for hidden features C = 256 for 120 epochs with a learning rate $1e-4$ and weight decay of $1e-8$.

## A.6 ADDITIONAL FIGURES

**Illustrating the effect of equivariant models in ShapeNet segmentation**

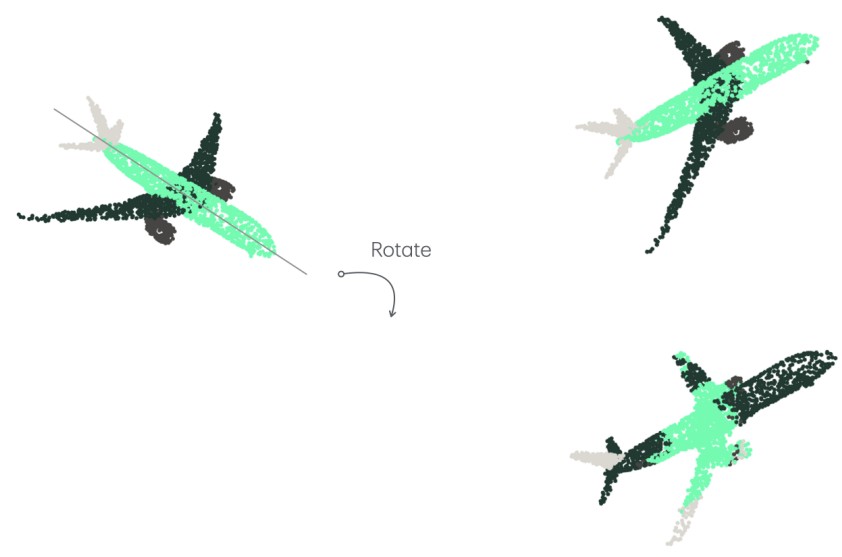

Figure 4: Depiction of an instance of ShapeNet segmentation for the class airplane. When the sample is rotated, a non-equivariant model mistakes wings for the nose of the airplane, while a rotationally equivariant model does the segmentation perfectly.

**Human motion prediction using CMU motion capture dataset**

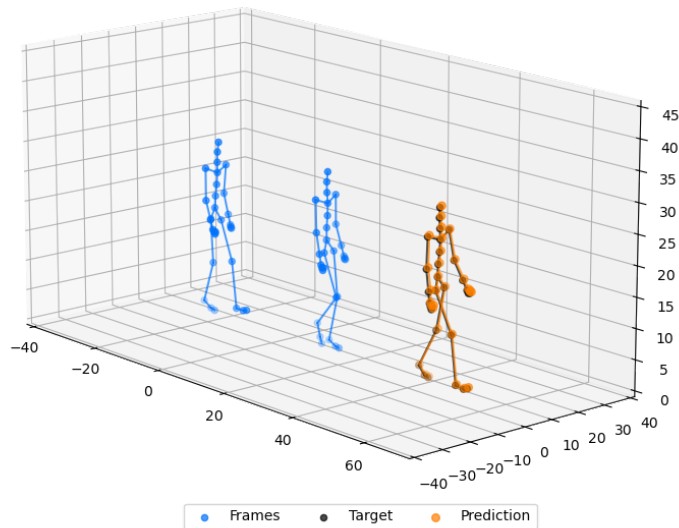

Figure 5: Depiction of an instance from CMU motion capture dataset.

