# OpenReview forum: "On the utility of Equivariance and Symmetry Breaking in Deep learning architectures on point clouds"
_ICLR.cc/2025/Conference — Submitted to ICLR 2025_

### Official Review · Reviewer_5SuB · 2024-11-01

**Soundness:** 2
**Presentation:** 3
**Contribution:** 2
**Rating:** 6
**Confidence:** 3

**Summary:**

This paper studies the impact of additional input information and SE(3)  equivariance on the performance of models processing point cloud data.  They present a series of hypotheses to  study different aspects of equivariant neural networks and symmetry breaking. Extensive experiments have been conducted on segmentation, regression, and generation tasks to verify the applicability and superiority of equivariant networks.

**Strengths:**

1. Detailed introduction to previous work.
2. The relationship between task complexity and equivariance was studied, and it was found that the equivariant method has more obvious advantages in tasks that require strict equivariance. It is demonstrated that providing explicit geometric information can improve performance even in the case of symmetry breaking.

**Weaknesses:**

1.There are so many contents from previous works that I can hardly tell the novelty. From my point of view, only Formulas 8 and 9 are new, but it is still quite easy to prove.

2.Although the paper proposes that symmetry breaking may be beneficial, it provides insufficient explanation of the mechanism behind it.

**Questions:**

None

---

> ### Author Response · Authors · 2024-11-21
> **Response to reviewer 5SuB**
>
> Dear reviewer, we thank you for your positive comments regarding the detailed background section and for recognizing the importance of the relationship between task complexity and equivariance. We also appreciate your constructive feedback and questions which we address as follows:
>
> 1. Regarding novelty: We agree that a better exposition of the novelty would improve the paper, as it may seem that we provide new theoretical results. Although we do not claim new theoretical results, we understand the confusion and will more explicitly state the experimental value of our paper.
>     * a. We primarily organize the results that the equivariant deep learning field has developed over the years and put them in a thorough benchmarking structure based on theoretically motivated hypotheses. We thereby aim to constructively contribute to debates surrounding the utility of equivariance in DL. This is our main contribution and we will emphasize it as such in the main paper.
>     * b. Additionally, as a contribution, we provide a new general-purpose architecture for point clouds, Rapidash, that allows for handling various forms of inputs (scalars and vectors) with multiple degrees of equivariance constraints.
>     * c. To further clarify Eqs 8 and 9. These are used to describe our Rapidash model. These are common definitions of group convolutions, but specifically, we follow the approach taken by (Bekkers et al, 2024) as cited in the paragraph preceding the equations.
> 2. We thank you for pointing out the insufficient explanation of the benefits of symmetry breaking. We have added the following text to the paper:
>     * a. From Tables 1, 2, and 3, we see that models that break SE3 symmetry perform well which supports the notion that having more information like a global pose, helps in performance despite not being exactly equivariant (Hypothesis 5). More explicitly, one can take a look at all the models that only have effective equivariance T3 or none, and compare them against their closest counterparts that have still SO(3) or SE(3) equivariance. For example in Table 1, one could compare model 7 (84.41%) to model 8 (85.60%). Both take normal vectors as input, but 8 additionally has access to a global pose which makes the model only conditionally invariant (the pose should rotate according to the point cloud). Or Compare model 7 (84.41%) to model 13 (85.41%), which both take the same amount of information as input, but 13 takes the normals as scalar inputs and this breaks invariance since then they are expressed in a global coordinate system. We will update the paper with additional details on symmetry breaking in the background section, and add an appendix with further clarifications. The results sections will more precisely refer to the results tables.

---

### Official Review · Reviewer_HwhT · 2024-11-03

**Soundness:** 3
**Presentation:** 3
**Contribution:** 3
**Rating:** 6
**Confidence:** 3

**Summary:**

This paper explores the key properties of deep learning networks on 3D point clouds, especially focusing on SE(3) equivariance. It conduct extensive experiments to test the initial hypothesis of the trade-offs between flexibility and weight-sharing introduced by equivariant layers. Based on the experimental results, a scalable network called Rapidash is introduced to facilitate the comprehensive testing of the hypothesis.

**Strengths:**

1. Clear motivation and insightful key idea.
2. The paper writing quality is very high, with explicit statements and clear logic links.
3. The code is provided in the supplementary material.
4. The analysis and theoretical conclusion may bring future insight to 3D point cloud learning.

**Weaknesses:**

1. The major concern lies in the dataset. ShapeNet is a synthetic dataset. However, one important key challenge of modern 3D point cloud networks is their real-world performance on real-world data. Thus, whether the raised hypothesis can also be accepted in real-world 3D data needs to be explored.
2. Although this paper verified the three important hypotheses, a **solution** derived from hypotheses such as some network design ideas or even *engineering tricks/techniques* (e.g., how to revise the current point convolution operation) should be further discussed and evaluated based on the extensive analysis of the paper, which will benefit the network design of future point cloud learning methods.
3. The reviewer believes that some extra figure illustrations should be provided to better intuitively understand the proposed theoretical hypotheses.

*Conclusion*:
The current version of the paper is theoretically important, but need more pilot studies to provide the **practical** values.

**Questions:**

1. Some pilot studies (no need too much) should be conducted on real-world datasets, or at least some robustness point cloud datasets. For example, ScanObjectNN for point cloud classification, or ModelNet40-C for robustness point cloud recognition.
2. Some "solutions" should also be provided and evaluated (some pilot studies are enough).

If these two questions can be addressed well, the reviewer may consider raising the rating.

---

> ### Author Response · Authors · 2024-11-21
> **Response to reviewer HwhT**
>
> Dear reviewer, we thank you for the time you put into reviewing our work and appreciate the positive remarks about the paper writing, motivation, and theoretical insights presented in the paper. Additionally, thank you for your constructive criticism, we answer the concerns and questions below:
>
> 1. We agree with your comment about the lack of practical implications and thus are working towards doing classification experiments on the ModelNet40-C dataset as suggested. We hope that the insights derived from these experiments can translate well to the real world. We will add the validity of our hypothesis to the discussion section as soon as the results come in.
> 2. We will expand our discussion section with practical implications following the hypothesis. To give an example, on Shapenet segmentation the gain in group convolution methods is negligible, but requires ~5x more compute. Then it is advised to use simple and fast non-equivariant methods. However, if equivariance is required for a task, equivariant architectures may be essential. See e.g. QM9, but also see Shapenet segmentation results which heavily drop for non-equivariant models when the input is not aligned with the axis).
> 3. Regarding additional explainer figures, we will add one that explains the utility of symmetry breaking when tasks do not strictly require this (s.a. shapenet segmentation).
>
> Answering the questions:
>
> * Q1. Taking note of this, we are working towards having a table for ModelNet40 C classification experiments to have a dataset that can represent real-world point clouds more accurately.
>
> * Q2. From the experiments, we plan to add the insights and validity of the presented hypotheses in the discussion section.

---

> > ### Comment · Reviewer_HwhT · 2024-11-25
> > **Keep the positive scores**
> >
> > After reading the rebuttal from authors and other reviews, I would keep my rating of Borderline Accept, towards a positive rating.

---

> > > ### Author Response · Authors · 2024-12-04
> > > **Updated pdf**
> > >
> > > Thank you for responding. Additionally, we have updated the pdf with initial experiments for ModelNet40 and ModelNet40C classification in the Appendix. We have added an initial analysis of the results of these experiments.

---

### Official Review · Reviewer_Wenf · 2024-11-03

**Soundness:** 2
**Presentation:** 2
**Contribution:** 2
**Rating:** 5
**Confidence:** 2

**Summary:**

Summary
This paper examines the role of equivariance and symmetry breaking in deep learning architectures for point clouds, focusing on the influence of additional input information and SE(3) equivariance on model performance. Through a series of experiments on various tasks and datasets (e.g., Shapenet 3D, QM9, and CMU Motion Capture), the authors compare the effects of equivariant and non-equivariant layers, exploring the advantages of equivariance as task complexity increases. Results show that equivariance offers significant benefits for small datasets and geometrically complex tasks, though this advantage diminishes in large-scale data regimes.

Strengths

Weaknesses

**Strengths:**

Novel Research Problem: The paper presents several hypotheses regarding the impact of equivariant vs. non-equivariant architectures, particularly on tasks of varying complexity. This is a novel and relevant problem, providing new insights into geometric deep learning for point cloud data.
Comprehensive Experimental Design: The authors conduct extensive experiments across multiple datasets, thoroughly evaluating the performance of different architectures, thus providing strong support for the effectiveness of equivariance.
Modular and Scalable Architecture: The proposed Rapidash architecture is extensible, supporting various forms of equivariance, and provides an efficient platform for testing, which is beneficial for future research.
Clear and Substantiated Conclusions: The experiments confirm the advantages of equivariance in geometrically complex tasks and quantify the impact on different data scales, offering practical guidance for model selection and design.

**Weaknesses:**

Limited Discussion on Symmetry Breaking: Although the paper addresses symmetry breaking by incorporating global coordinates, it lacks a detailed analysis of how this affects performance across different tasks. Additional theoretical insights or quantitative results would strengthen this discussion.
Insufficient Evaluation of Model Complexity and Computation Costs: While Rapidash shows promise for high-complexity tasks, the paper provides limited discussion on computational costs and scalability in practical applications, which may impact its usability for large-scale deployment.
Limited Benchmark Comparisons: While some baseline methods are included, the paper does not cover all recent benchmarks. Broader comparisons with state-of-the-art equivariant and non-equivariant methods would enhance the persuasiveness of the findings.
Lack of Practical Application Discussion: Although theoretically relevant, the paper does not discuss the potential impact of equivariance and symmetry breaking on real-world point cloud applications (e.g., 3D reconstruction or object detection), potentially limiting its practical value.

**Questions:**

see Weaknesses

---

> ### Author Response · Authors · 2024-11-21
> **Response**
>
> Dear reviewer, we appreciate the time taken to review our work and for carefully going through the text and giving constructive feedback. We are grateful for your feedback and will answer the concerns/questions mentioned in the review, below:
>
> 1. Limited discussion on symmetry breaking: We appreciate that you pointed out that the current version of the paper does not clearly emphasize the importance of symmetry breaking as a clarification would greatly improve the paper. We address this issue by adding details in an additional appendix and in the background section (an updated pdf will follow shortly):
>    * a. In short, we now distinguish two types of symmetry breaking: **internal** and **external**. With **internal** we mean that if the layers are intrinsically equivariant (which ours are) then the layers themselves can be made non-equivariant (symmetry breaking) by relaxing equivariance constraints, recent literature explored such type of symmetry breaking [1, 2, 3]. By **external** we mean that architectures—that are intrinsically equivariant—can be broken to be non-equivariant by external factors, such as providing inputs that themselves are not invariants/covariants. For example, when providing input coordinates as triplets of scalars in a global reference frame, then this breaks equivariance as now we have an orientation/pose-dependent input. When providing coordinates as vectors, however, the inputs will internally be described relative to local reference frames (the orientation grids) and symmetry is preserved.
>     * b. We further clarify that in contrast to related works on symmetry breaking, our study primarily focuses on external symmetry breaking (cf the red exclamation marks in the tables).
>     * c. We will add an explanation to the main text, and include the following references for the context of internal symmetry breaking:
>         - [1] Wang, R., Walters, R., & Yu, R. (2022, June). Approximately equivariant networks for imperfectly symmetric dynamics. In International Conference on Machine Learning (pp. 23078-23091). PMLR.
>         - [2] van der Ouderaa, T., Romero, D. W., & van der Wilk, M. (2022). Relaxing equivariance constraints with non-stationary continuous filters. *Advances in Neural Information Processing Systems*, *35*, 33818-33830
>         - [3] Kim, H., Lee, H., Yang, H., & Lee, J. (2023, July). Regularizing towards soft equivariance under mixed symmetries. In *International Conference on Machine Learning* (pp. 16712-16727). PMLR.
> 2. Insufficient Evaluation of Model Complexity and Computation Costs: We are working towards classification experiments on ModelNet40-C and plan to add the findings to the discussion section. We hope that adds to the understanding of how these models perform on real-world-like data. Regarding computational complexity, we already reported epoch time per model, but now further discuss in the main text that for non-equivariant tasks (such as e.g. Shapenet segmentation), increased computational costs of group convolutional methods (~5x slower) do not outweigh performance gains. In strictly equivariant tasks (such as QM9), the performance improvement is rather significant in which case it would outweigh the costs.
> 3. Limited Benchmark Comparisons: The decision to add fewer benchmarks was purely made due to space constraints. To address this, we will keep a short list of reference methods in the current tables but additionally provide separate tables for an exhaustive comparison to the literature in the appendix. Additionally, we will cite the relevant papers in the main text associated with each experiment. We hope this answers your concerns.
> 4. Lack of Practical Application Discussion: We thank you for bringing this to our attention and we agree that the practical applications of this work are not thoroughly discussed. We will expand our discussion section and are currently running experiments on ModelNet40C to gain additional insights towards this end.

---

### Official Review · Reviewer_4wHY · 2024-11-04

**Soundness:** 3
**Presentation:** 3
**Contribution:** 3
**Rating:** 5
**Confidence:** 4

**Summary:**

This work investigates the utility of imposing equivariant constraints in models that perform point cloud processing. To study the effects of the application of different equivariant constraints, the authors propose a scalable architectural desing that allows for easily incorporating varying degrees of equivariance. They evaluate this architecture on different point cloud processing tasks, incorporating different levels of equivariance or symmetry-breaking factors that break the network's symmetry constraint. Experimental results are used to accept or reject a set of hypotheses regarding the effects of equivariant constraints on model performance and generalization. Specifically, this work analyzes the effects of equivariance when modifying the size of the model, the size of the dataset, the complexity of the task, or when allowing symmetry breaking by providing pose dependent information (that is not available in the typical equivariance case).

**Strengths:**

- This work provides a detailed description of the most commonly used equivariant architectures for point-cloud processing.
- Additionally, the clear definition of the hypotheses that the authors aim to investigate allows for the explicit design of experiments that can be used to test them, facilitating the reader's understanding of the conclusions drawn from the experimental observations.
- The extensive experimental evaluation across diverse point-cloud tasks provides convincing evidence of the effect of difference levels of equivariance on the proposed Rapidash model, showing the influence of the dataset size and task complexity on these effects.

**Weaknesses:**

- This work lacks sufficient attribution to prior work that has investigated hypotheses similar to the ones tested here. For example, hypothesis 5, on the effect of symmetry breaking in equivariant neural networks, has already been studied in prior works, such as :

	[1] Marc Finzi, Gregory Benton, Andrew Gordon Wilson "Residual Pathway Priors for Soft Equivariance Constraints"

	[2]  Mircea Petrache, Shubhendu Trivedi, "Approximation-Generalization Trade-offs under (Approximate) Group Equivariance"

	[3] Stefanos Pertigkiozoglou, Evangelos Chatzipantazis, Shubhendu Trivedi, Kostas Daniilidis , "Improving Equivariant Model Training via Constraint Relaxation"
- The experimental evaluation is limited to examining the effects of the equivariant constraints on a single architecture. While it is shown that these effects generalize across tasks it is not clear how they generalize across different model architectures.
- In the experimental results, no error bars are provided. As a result, it is hard to assess the significance of the observations that are used to accept or reject the various hypotheses.

**Questions:**

- Providing a more detailed discussion on how the proposed hypotheses have also been investigated in prior works would improve the completeness of this work.
- Why is it reasonable to expect that the experimental observations generalize across different model architectures?
- Including error bars and variances of the reported numbers, given the randomness of the model's initialization and training, would allow for easier verification of the significance of the reported observations.

---

> ### Author Response · Authors · 2024-11-21
> **Response**
>
> Dear reviewer, We thank you for the time and effort you have put into providing a detailed review of our work. We value your constructive feedback and try our best to address your concerns and questions below:
>
> 1. On prior work: We recognize this as an error on our part and thank the reviewer for listing important papers that we missed. We have updated our background section with suggested and additional citations, and added a short appendix that explains our view on symmetry breaking, see also our response to the reviewer regarding symmetry breaking [Wenf].
> 2. On the generalization of results beyond convolutional architectures: Thank you for raising this important point. We agree that demonstrating generalization across different architectures would strengthen our findings, and we appreciate the opportunity to clarify our approach and expand our discussion on this topic.
>    * a. To systematically evaluate our hypotheses, we focused on a single architectural class that enables precise control over equivariance constraints. This allowed us to conduct detailed ablation studies across 58 model variations and five tasks, isolating the effects of varying equivariance constraints on model performance.
>     * b. While the literature includes various equivariant architectures (from convolutional to transformer-based), Rapidash's unique flexibility in handling different forms of equivariance constraints was essential for our systematic study. Extending this analysis to other architectures, such as transformers, would require developing a new model class with similar flexibility - an interesting direction beyond our current scope.
>     * c. We believe our findings about equivariance effects should in principle generalize across model classes since our ablations isolate fundamental properties of equivariance constraints. *We’ll add a concise discussion to the paper*, referencing several other works (including those on equivariant transformers) that suggest our findings might generalize, such as e.g. EquiformerV2 [1] which shows that fewer constraints (higher degree irreps) improve performance on equivariant tasks (our hypothesis 4). Also, that paper shows that for strictly equivariant tasks, symmetry breaking is not beneficial, which is something we also observe.
>     * d. *However, we are very interested in your perspective on this*. Are there specific mechanisms that might cause different behaviors in other architectures that prevent generalization of our results? Either way, in the text, we will delineate the scope of our findings to be limited to the convolutional class.
>     * e. Finally, to facilitate future work in the direction of generalizing our findings, we have provided our complete codebase and benchmark implementations in the supplementary materials (anonymized .zip file; upon acceptance, we will publish our code in a public repo). This enables direct verification and extension of our findings to other architectural classes.
> 3. Missing error bars: We agree this is necessary and we are currently running our models with multiple seeds and will update the tables and pdf shortly.
> - Q1: We plan to update our paper with additional citations in the related work and discussion.
> - Q2: See our response above. We consider doing analysis for different architecture types to be beyond the scope of this work and emphasize that these results hold within the convolutional neural network class paradigm. We agree that the generalizability of the hypotheses remains to be tested outside of the discussed architecture.
> - Q3: We are currently collecting those results and they will be uploaded with the updated pdf in due time.
>
> [1]: EquiformerV2: Improved Equivariant Transformer for Scaling to Higher-Degree Representations, Liao et al, ICLR 2024.

---

### Author Response · Authors · 2024-12-04
**Summary of rebuttal period**

Dear reviewers and ACs

Thank you for your invaluable contributions to the review process. We are grateful for the time, effort, and feedback provided by the reviewers.

We next address general points raised jointly among the reviewers and proceed to respond to specific comments made by reviewers. Firstly, we are glad that you found the problem at hand **novel problem** and found our approach **providing new insights** & **insightful key idea** (reviewer Wenf, reviewer HwhT). The proposed architecture in this work, Rapidash was found useful to conduct the experiments: **is extensible, supporting various forms of equivariance, and provides an efficient platform for testing, which is beneficial for future research** (reviewer Wenf). Our experimental analysis was well-received and reviewers found it **extensive experimental evaluation across diverse point-cloud tasks, extensive experiments across multiple datasets, thoroughly evaluating the performance** (reviewer 4wHY, Wenf ).

The reviewers identify valuable points for improvement which we have incorporated in the updated version of the paper. The main additions and experiments in response to reviewers’ concerns include the following:

1. *Regarding lack of error bars as mentioned by reviewer 4wHY -* We have added variances to all the results presented in Table 1 and Table 2, as well as the best for the performing models in Table 3. ( We plan to add the rest of the results in Table 3 in the camera-ready version of the paper as they did not finish within the rebuttal pdf time period)
2. *Regarding the lack of attribution to related work (reviewer 4wHY) and limited discussion on symmetry breaking (reviewer Wenf, reviewer 4wHY),* we have added the suggested related work and more to the paper's main text. An additional paragraph discussing symmetry breaking is added in section 3.1 and Appendix 2.
3. *Limited benchmark comparisons (reviewer Wenf),* We have added additional benchmark results to Table 1, 2 and 3.
4. *Lack of practical application discussion (reviewer Wenf, reviewer HwhT):* We have added additional experiments for classification on ModelNet40 and ModelNet40C datasets, as well as a discussion section discussing the findings based on the two datasets in the Appendix.
5. *Clarity on contribution (reviewer 5SuB) and additional figure (reviewer HwhT)* we have updated our contributions in the main text and added an illustration of equivariant vs non-equivariant models on the segmentation task in the Appendix.

We have addressed each of the reviewers' concerns and questions at length as a response to the reviews. We note that while we were unable to receive further feedback from 3 out of 4 reviewers regarding our implemented improvements, we are confident that the comprehensive additions and clarifications we have provided directly address the concerns raised in their initial reviews. We believe these substantial improvements strengthen the paper considerably and hope they will be taken into account in the final decision process.

Best
Authors

---

### Meta-Review · Area_Chair_Bpsa · 2024-12-23

**Metareview:**

This paper investigates the utility of imposing equivariant constraints in models that perform point cloud processing. The manuscript was reviewed by four experts in the field. The recommendations are (2 x "5: marginally below the acceptance threshold", 2 x "6: marginally above the acceptance threshold"). The reviewers raised many concerns regarding the paper, e.g., limited technical novelty, unclear motivation and statement, unconvincing experimental evaluations, missing comparisons with previous literature, etc. Considering the reviewers' concerns, we regret that the paper cannot be recommended for acceptance at this time. The authors are encouraged to consider the reviewers' comments when revising the paper for submission elsewhere.

**Additional Comments On Reviewer Discussion:**

Reviewers mainly hold concerns regarding limited technical novelty (e.g., solutions seem to be engineering tricks/techniques, many contents are from previous works) (Reviewer HwhT, 5SuB), unclear motivation and statement (e.g., lack of in-depth and detailed analysis of symmetry breaking, no theoretical insights, insufficient explanation) (Reviewer Wenf, 5SuB), unconvincing experimental evaluations (e.g., limited on a single architecture, no error bars, no computational costs and scalability, synthetic dataset is not enough) (Reviewer 4wHY, Wenf, HwhT), missing comparisons with previous literature (e.g., hypothesis 5 similar to existing works, no recent benchmarks and methods) (Reviewer 4wHY, Wenf). The authors' rebuttal could not fully address the above-listed concerns.

---

### Decision · Program_Chairs · 2025-01-22

Reject